# Exhaled Biomarkers for Point-of-Care Diagnosis: Recent Advances and New Challenges in Breathomics

**DOI:** 10.3390/mi14020391

**Published:** 2023-02-04

**Authors:** Helga Kiss, Zoltán Örlős, Áron Gellért, Zsolt Megyesfalvi, Angéla Mikáczó, Anna Sárközi, Attila Vaskó, Zsuzsanna Miklós, Ildikó Horváth

**Affiliations:** 1National Koranyi Institute for Pulmonology, Koranyi F Street 1, 1121 Budapest, Hungary; 2Department of Pulmonology, University of Debrecen, Nagyerdei krt 98, 4032 Debrecen, Hungary

**Keywords:** exhaled nitric oxide, exhaled carbon monoxide, exhaled hydrogen sulfide, electronic nose, volatile organic compounds, biosensors, breathomics, artificial olfaction system, COVID-19

## Abstract

Cancers, chronic diseases and respiratory infections are major causes of mortality and present diagnostic and therapeutic challenges for health care. There is an unmet medical need for non-invasive, easy-to-use biomarkers for the early diagnosis, phenotyping, predicting and monitoring of the therapeutic responses of these disorders. Exhaled breath sampling is an attractive choice that has gained attention in recent years. Exhaled nitric oxide measurement used as a predictive biomarker of the response to anti-eosinophil therapy in severe asthma has paved the way for other exhaled breath biomarkers. Advances in laser and nanosensor technologies and spectrometry together with widespread use of algorithms and artificial intelligence have facilitated research on volatile organic compounds and artificial olfaction systems to develop new exhaled biomarkers. We aim to provide an overview of the recent advances in and challenges of exhaled biomarker measurements with an emphasis on the applicability of their measurement as a non-invasive, point-of-care diagnostic and monitoring tool.

## 1. Introduction

The lung is an important interphase between the environment and the human body, and it serves as a major getaway for different biomolecules. Complex biological processes in different body organs have their fingerprints on exhaled breath by releasing gas phase mediators and other biomolecules that are transported to the lungs and released into the exhaled breath through the alveoli. The lung parenchyma and the airways are major sources of mediators released to the airways and make a substantial contribution to the content of exhaled breath.

### 1.1. The Path of Using Exhaled Volatile Compounds in Medicine

The potential of using exhaled breath to obtain information about different body functions was first recognized at the time of ancient Greek medicine when special odors were linked with different diseases such as liver cirrhosis and diabetes. It took centuries to identify and quantify the biomolecules responsible for the signals sensed by human olfaction. A landmark study was published by Pauling L et al. [1] in 1971 demonstrating the presence of hundreds of volatiles in exhaled breath samples using gas–liquid partition chromatography. With the advent of gas chromatography and mass spectrometry researchers have identified and quantified thousands of volatile organic compounds (VOCs) in the breath, most of them in picomolar (10–12 mol/L or particles per trillion) concentrations [2,3,4]. Different diseases have characteristic metabolic profiles that can be captured by using exhaled VOC profiles (“breathprints”). For the interpretation of huge datasets arising from a complex mixture of thousands of widely different volatile molecules to provide clinically relevant information for discrimination between health and disease and for the prediction of therapeutical responses, several statistical algorithms have been used resulting in variable levels of diagnostic accuracy [5,6]. The large size of mass spectrometers, and the substantial expense and heavy workload required for sample processing have represented a major bottleneck for the point-of-care (POC) clinical applicability of these measurements. Two small molecules, hydrogen (H_2_) and methane (CH_4_), represent good examples of this transition, as they have made their way to be measured by POC tests and are widely used in the differential diagnosis of gastrointestinal disorders [7,8]. Hydrogen and methane-based breath tests are used to diagnose and monitor small intestinal bacterial overgrowth and carbohydrate maldigestion and guide clinicians to prescribe appropriate medication [9,10]. These tests are based on the observation that H_2_ and CH_4_ are produced by the bacterial fermentation of unabsorbed carbohydrate in the small intestine during digestion and diffused to the blood that carries them to the alveoli from where they are exhaled. Since human cells do not produce them, their concentrations in breath are related to the interstitial bacterial flora.

### 1.2. Gaso-Transmitters in Exhaled Breath

As well as VOCs, the environmental-pollutant-free radical nitric oxide (NO), a known gaso-transmitter in the body, was also detected in exhaled breath with trace concentrations in healthy subjects and elevated levels in asthmatic patients [11,12]. Determination of fractional exhaled NO (FeNO) has generated great interest as a potential biomarker of asthma. This was mainly based on its correlation with eosinophils and its increase after allergen exposure, suggesting that it may be useful as a predictive marker of asthma attacks and the therapeutic response [13,14,15]. FeNO has served as a prototype of exhaled biomarkers for disease monitoring and medical decision making. Several machines have US Food and Drug Administration approval and/or a European Union CE-mark as medical device for its measurement [16]. The other two toxic environmental pollutants with known gaso-transmitter functions in the human body, carbon monoxide (CO) and hydrogen sulfide (H_2_S), can also be detected in exhaled breath. Their levels are altered in different diseases, such as asthma, chronic obstructive pulmonary disease (COPD) and cystic fibrosis [17,18,19,20,21]. However, the use of exhaled CO as a biomarker of heme oxygenase activity is hampered by the strong and long-standing effect of smoking on the exhaled CO level, and the exhaled H_2_S level is profoundly influenced by its oral and gastrointestinal bacterial production [20,22].

### 1.3. Biological and Artificial Olfaction Systems to Assess Exhaled Volatiles

Several species have a lot more sensitive olfactory systems than humans including dogs, rats and different insects. The specific coupling of large numbers of receptors with the brain neural network enables these species to recognize minor changes in the volatome of different human samples including the breath. This led to the idea of involving trained animals in human medicine and diagnostics. Sniffer dogs have been trained successfully to distinguish biological samples obtained from healthy and diseased individuals. They have been shown to identify patients with Parkinson’s disease [23], lung cancer [24], prostate cancer [25], ovarian cancer [26] and different infectious diseases [27] from samples such as urine, blood, serum, cell lines and bacterial cultures with very high sensitivity. Dogs can also be trained to alert to hypoglycemic periods in type 1 diabetics [28]. Moreover, even untrained dogs have been shown to sniff out the prodromal phase of seizures and respond to the unusual odor changes with an increase in affiliative behaviour directed at their owners [29]. As well as dogs, other animal species have been tested in odor-pattern-based diagnostics. For instance, African giant pouched rats can detect *Mycobacterium tuberculosis*, the pathogens causing tuberculosis very sensitively [30,31], and are indeed used for first line diagnostics in Africa. Insects, such as mosquitos and honeybees also have a very sensitive olfaction with great discriminatory power to detect a tremendous amount of chemical signals [32,33]. Moreover, bees have already been successfully trained to detect specific odors [34].

Compared to the detection and quantification of individual molecules, using the biological olfactory systems of animals as a model to build artificial olfaction systems, so-called electronic noses, is a completely different approach. Electronic noses consist of arrays of chemical vapor sensors that respond to certain characteristics of odorant molecules including exhaled VOCs. Sensors are not specific to a given molecule, a sensor may react with several different molecules and a given molecule may also generate responses from several sensors. In this approach individual molecules are not identified and quantified as they are by mass spectrometry; only the pattern of sensor responses (“breathprint”) induced by a complex mixture of different volatiles is clustered. Despite the limitation of the black box approach due to the versatile nature of potential arrays of chemosensitive sensors, their small size and low cost, they have gained great attention as potential point-of-care clinical tools [35,36,37]. Their integration with artificial intelligence for data analysis has contributed importantly to the rapid development of this field [38].

### 1.4. Methodological Issues Related to Breath Sampling

Breath samples are easily accessible; however, samples vary from breath to breath, and there are several methodological issues that make data comparison difficult between different laboratories, therefore limiting clinical utility. Recognition of the need for standardization has resulted in a large number of methodological studies providing support for the development of exhaled biomarkers for clinical use. Different research groups have addressed areas such as environmental air, breathing pattern, types of exhalation, nasal and oral influence, smoking, food, circadian rhythm, medication, together with collecting time, temperature, humidity, sampling bags, storage and analytic methods to clarify their influence on exhaled biomarker levels. The European Respiratory Society (ERS) and the American Thoracic Society (ATS) have published guidelines and recommendations for exhaled biomarker measurements starting from exhaled nitric oxide determination to the standardized sampling and measurement of VOCs [39,40,41].

We aim to provide a comprehensive review on the measurement of exhaled gaso-transmitters and VOCs with a special focus on their potential use as a point-of-care tool in clinical practice. We also highlight areas where further development is needed.

## 2. Exhaled Gaso-Transmitters

There are three known gaso-transmitters in the human body: NO, CO and H_2_S. They are widely different molecules. They are all counted as environmental pollutants and toxic gases. As bioactive molecules they have important anti-inflammatory, antioxidative, antiproliferative and antiapoptotic properties, and their low-dose inhalation or administration of their donor molecules can provide therapeutic effects in different conditions [42,43,44]. Due to their environmental occurrence, when their levels are measured in exhaled breath, special attention is required to limit the potential environmental influence. This is a complex task because it is not enough to determine the background environmental levels as environmental gases once inhaled could stay in the human body for different time lengths that depends on their physicochemical nature. They either could be exhaled immediately, or they might pass the alveolo-capillary membranes and circulate in the body for several hours and be added to exhaled breath in later breathing cycles [22,39]. They interact with different molecules, and in this way, they can be transformed into other molecules that may result in lower than environmental concentrations in exhaled breath. The other methodological challenge is that their bodily production and transportation results in very low concentrations being present in exhaled breath, requiring very sensitive detection systems.

### 2.1. Exhaled Nitric Oxide

NO was first described as a signaling molecule in the vasculature in the late 1980s. In 1998, three eminent researchers, Furchgott RF, Ignarro LJ and Murad F, won the Nobel price for their discoveries of the signaling function of NO, a short-lived gas molecule, which was a completely new principle for communication between the cells in the human body [45,46]. NO is the oxidation product of L-arginine released during the reaction catalyzed by nitric oxide synthase (NOS). Different NOS isoforms have been identified, with the constitutive ones (neuronal and endothelial NOS) being expressed in the epithelium, endothelium, platelets, neural tissues and skeletal muscles. They play a role in synaptic plasticity, blood pressure regulation, neurovascular coupling, smooth muscle relaxation, penile erection and anti-atherosclerotic and other vasoprotective effects [47,48,49]. The inducible form (iNOS) is upregulated by bacterial lipopolysaccharide and several inflammatory cytokines in a variety of cells and produces a large amount of NO. NO overproduction frequently occurs at sites of inflammation where the production of reactive oxygen species (ROS) including superoxide (O_2_^−^) and hydrogen peroxide (H_2_O_2_) is also enhanced [50]. The interaction between the two free radicals, NO and O_2_^−^, results in the formation of more oxidative products, such as peroxynitrite, while NO without superoxide yields N-nitrosamine derivatives. NO therefore plays a dual role: it has a protective effect in non-specific immune defense, and it also has toxic and pro-inflammatory effects and mediates various symptoms of inflammation, oxidative stress and septic shock [51]. There are several therapeutic applications of the modulation of NOS pathways and the direct administration of low-dose NO including the treatment of hypertension, peripheral arterial disease, sepsis and acute respiratory failure [51,52,53,54,55].

In the airways, iNOS is located in the airway epithelium, mainly in the larger airways. In asthmatic patients, overexpression of airway iNOS caused by type 2 cytokine interleukin-13 (IL-13) is a major source of an increased level of exhaled NO. Treatment with steroids inhibits the upregulation and results in a decrease in the exhaled NO level [51]. Furthermore, the inhibition of iNOS causes a decrease in the FeNO level of asthmatic patients [56].

NO is present in trace amounts in exhaled breath [11,12,13,14,15,40]. Originally, highly sensitive chemiluminescence analyzers used for environmental NO detection were adapted for FeNO measurement [11,12,39] (Figure 1).

These machines use chemiluminescence for NO detection. Exhaled nitric oxide reacts with ozone (O_3_) releasing photons that are captured by a photomultiplier. The number of photons is proportional to the nitric oxide concentration and visualized in real time. These types of equipment are sensitive to 0.1–0.5 parts per billion (10^−9^ ppb) with good reproducibility features and a fast response time (0.5–0.7 s). A unit is added to the analyzers to control expiratory flow. The ozone generator is included in the equipment. The technique requires frequent calibration for everyday use and a yearly technical service. As early observations showed that the fractioned exhaled nitric oxide (FeNO) level depends on the exhaled air flow rates, the international research community made a great effort to standardize exhalation techniques to enable measurement sites to compare their data [39,40]. Through this process, the methodology of FeNO measurement has evolved from measuring NO during uncontrolled breathing maneuvers to well-standardized sampling. Generally, a forced exhalation from total (or close to total) lung capacity (TLC) with a set slow expiratory flow rate (45–55 mL·s^−1^) against resistance (5–20 cm H_2_O) is able to close the velum to exclude influence from high NO levels of the upper airways and nasal cavities. The measurement is repeated twice, and if values of the two plateaus are within 10% of each other, the measurement is technically acceptable. The FeNO level is taken at the end of the plateau phase of the measurement. There are different technical solutions to minimize the effect of environmental NO, which can reach values in the range of 50–100 ppb in polluted areas, i.e., much higher than exhaled NO levels in healthy subjects (median of 16.5 ppb). These include inhaling NO-free gas or using an NO “scrubber”.

Breath samples for FeNO measurement can be collected both online and offline. For offline sampling, the same exhalation maneuver should be performed. To exclude upper airway contamination, it is necessary to exclude dead space gas either by discarding the first 150–200 mL of a sample or start sampling only after the appearance of the CO_2_ signal.

The ATS and ERS guidelines recommend an expiratory flow rate of 50 mL/s for the use of FeNO in clinical practice to provide the best discriminative power for steroid responsiveness in asthma (FeNO_50_). Several studies aimed at setting reference values for FeNO_50_. The study of Toren K et al. demonstrated that FeNO_50_ levels are significantly influenced by sex, height, age and atopy [40,57,58,59,60,61,62] and argued for individual reference values similar to those used in the lung function test. Furthermore, values were different between smokers and non-smokers. The median value of FeNO_50_ in female never smokers was 15.7 ppb and in males 19.0 ppb, while in smokers these values were 10.4 ppb and 13.2 ppb, respectively. In asthmatic patients, values can be higher than 100 ppb and show great variability [63], especially during exacerbation. In clinical practice, FeNO_50_ measurement has been used in asthma as a surrogate marker of eosinophilic airway inflammation and as a predictive biomarker of corticosteroid responsiveness [16,41,64]. Furthermore, its usefulness in specific conditions including pregnancy and smoking has also been clarified and uncertainties have also been addressed [65,66,67,68]. In general, a FeNO_50_ level >50 ppb is used as a predictor of a good response to steroid treatment in asthmatic patients. Values between 25–50 ppb are taken as indicators for a potential response and <25 ppb as no response (normal). However, the ATS guideline states that its experts did not find enough evidence to make cut-off values. They suggest using FeNO in combination with other measures of asthma control and that the level of FENO should be interpreted keeping in mind the given pretest probability [41].

Busse WW et al. demonstrated that FeNO >50 ppb is a good predictor of exacerbation even in patients with uncontrolled moderate-to-severe asthma [69]. There is an increasing use of FeNO as a biomarker of type-2 inflammation in severe asthma [70]. In patients with severe asthma, high levels of FeNO can be used to initiate anti-T2 biological treatment [69,71,72]. These treatment options result in a variable degree of decrease in FeNO [73].

In addition to asthma, steroids are also frequently used in chronic obstructive pulmonary disease (COPD), especially during exacerbations. Antus B et al. demonstrated that FeNO_50_ has a good predictive value for airway eosinophilia in acute exacerbation of COPD but not in stable disease [74,75].

Alterations in FeNO levels have been detected in several other diseases (Table 1). Due to the day-to-day variation in FeNO values and the confounding effects of different medications, the issue of individual baseline values and serial measurements are emphasized in areas where disease-related changes are not so profound as in asthma [76].

An endogenous inhibitor of nitric oxide synthase (eNOS), asymmetrical dimethylarginine (ADMA), is associated with a decreased bioavailability of NO and a lower level of FeNO and is a predictor of mortality in critical illness [77,78]. Changes in the ADMA level may also play a role in COPD and other respiratory disorders and may modify the FeNO level; however, this needs further study [79].

Since NO measurement has provided information on airway inflammation non-invasively and immediately after measurement, several groups have searched for solutions to provide equipment for POC service with small, cheap, easy to handle devices with low running costs.

**Table 1 micromachines-14-00391-t001:** FeNO alterations in different conditions and diseases.

Conditions/Diseases	FeNO Levels	References
Physical exercise	↓	[80,81]
Pregnancy	→	[67,68,82]
Smoking	↓	[66,83,84,85,86,87,88]
Pulmonary diseases		
Stable COPD	→↑	[89,90,91,92,93,94]
Severe COPD	↓	[95]
COPD exacerbation	↑	[96]
Cystic fibrosis	↓	[97,98]
Primary ciliary dyskinesia	↓	[99,100]
Bronchial asthma (esp. with eosinophilic airway inflammation)	↑	[65,66,68,82,101,102]
Interstitial lung disease	↑	[103]
Lung cancer	↑	[104,105,106]
Pulmonary tuberculosis	→	[107]
Pulmonary infections after lung transplantation	↑	[108]
Asbestos-related diseases	↑	[109,110]
Cardiovascular diseases		
Heart failure	↑	[111]
Atherosclerotic risk	↓	[112]
Pulmonary hypertension	↓	[113,114]
COVID-19 infection (CONTROVERSIAL!)		
Acute severe infection	↓	[115,116]
Lung parenchimal involvement, prognosis	↑	[117,118]
Post-COVID syndrome	↓	[115]
Inflammatory diseases		
Psoriasis, psoriatic arthritis	↑	[119,120]
Systemic sclerosis	↑	[121]
Inflammatory bowel disease	→	[122,123]
(Eosinophilic) esophagitis	→	[124,125]

To provide the clinically required level of discrimination between responders and non-responders to anti-eosinophil treatment in asthma, measurement techniques less sensitive and reproducible than chemiluminescence have been tried, and small, handheld, portable devices have been developed and compared with the stationary chemiluminescent analyzer and with each other [40,57,58,59,60,61,62]. These machines use electrochemical or optical techniques. For electrochemical detection, the main principle is based on the amperometric technique. The sensor transforms gas concentration to a detectable electrical signal (current). In detail, a buffer system causes retention of the last part of the exhaled sample in the instrument. The sample is transferred to an active catalytic sensor, where it undergoes a chemical reaction, and a physical change is emitted within an electrical circuit that is measurable. The signal is directly proportional to the partial pressure of NO in the sample. Electrochemical instruments are less sensitive than chemiluminescence detectors (5 ppb) with an accuracy of about ±5 ppb. The response time is longer (<10 s) and the analysis is slower (60–100 s) in portable machines than in stationary ones. Each device has a replaceable sensor that needs changing after a certain number of measurements (50; 100; 300) or after a certain time period (1–2 years). When a sensor is replaced, the performance indicators might change in these portable devices. As well as the disposable sensor cartridges, they usually use a built-in flow control biofeedback system and a processor with a dedicated software to visualize nitric oxide concentration in parts per billion (ppb). Between two measurements, at least 30 s tidal breathing is required. They are relatively cheap, but their running costs need to be considered [40]. Comparisons between different FeNO analyzers demonstrated reasonable reproducibility supporting the use of handheld devices in clinical practice [57,58,59,60,61,62]. However, these devices are not completely interchangeable and the ERS Task Force recommended they should not be changed in longitudinal studies.

In optical devices, exhaled NO interacts with laser-generated light causing a change in its intensity or polarization that is detected by a photodetector [40]. For NO detection in exhaled breath containing other molecules, the light source is required to be able to emit light in a well-specified narrow spectral range (5.1 to 5.7 μm) to limit interaction with molecules other than NO. Lasers able to produce such strictly specified wavelengths are, for example, the laser diodes (for example: tunable diode laser absorption spectroscopy /TDLAS/) [126]. They are sensitive in the low ppb range and can also detect CO and carbon dioxide (CO_2_). The latter is useful for monitoring exhalation while measuring NO. Another laser technology is the quantum cascade laser (QCL) that has also been assessed for exhaled NO measurement with different spectroscopies for signal detection [127]. Several developments were made in the last decades both for the light source (high power at room temperature; wavelength modulation) and signal detector (applying multi-pass cell configuration or high-finesse cavities in spectroscopy) enabling the technique to capture NO in a sub-ppb range. Developments in optical sensing based on absorption spectroscopy applying multi-pass cell configuration or high-finesse cavity (cavity-enhanced, cavity ring-down or photoacoustic spectroscopy) have further improved sensitivity [127,128,129,130]. Mandon J. et al. demonstrated that QCL can provide comparable results for FeNO monitoring to chemiluminescence and electrochemical methods [130]. Petralia L. S. et al. measured the temporal profile of exhaled NO and CO_2_ concentrations by the combination of optical sensors (diode laser absorption spectroscopy for at-mouth CO_2_ and quantum cascade laser-based, cavity-enhanced absorption cell for NO and side-stream CO_2_) to produce NO expirograms (F_E_NOgrams) to localize airway inflammation [131].

FeNO_50_ (and FeNO values at other low flow rates) represents NO release in the large airways. Changes in NO dynamics in the small airways and lung parenchyma requires extended NO measurement [40]. Extended nitric oxide (NO) analysis is a modelling tool for the partitioned measurement of nitrative stress in the conducting bronchi and peripheral airways/alveolar spaces. For this, mathematical modelling is used. The most used lung model divides the airways and lung into two compartments: a cylindrical tube represents the trachea and the large central airways (airway compartment) and an expansible part represents the respiratory bronchioli and alveoli (alveolar and acinar compartment) [40,132,133,134,135]. The trumpet-shape of the airways is also taken into account together with potential axial back diffusion of NO. A constant NO release to airspaces is assumed for both compartments.

For the modelling of NO dynamics at the central and peripheral parts, exhaled NO needs to be measured at least at two different flow rates (≥100 mL·s^−1^). Lazar Z et al. measured exhaled NO at constant flow rates of 50 mL s^−1^ (for FeNO_50_) and 100–150–200–250 mL s^−1^ for the extended NO analysis and created a suitable protocol for alveolar NO measurement [132]. The alveolar NO (C_A_NO) was calculated with the linear method from NO levels measured between 100 and 250 mL s^−1^ and was proved to be feasible even in patients with severe airflow obstruction. In asthma, C_A_NO can detect small airway involvement that cannot be determined by measuring FeNO_50_. In interstitial lung diseases, C_A_NO might reflect fibrogenesis and collagen deposition in the distal airspaces and is expected to be useful for clinical decision making [133]. Bronchial wall NO concentration (C_aw_NO) and the bronchial diffusivity of NO (D_aw_NO) can also be calculated by the extended NO analysis, providing detailed information about NO dynamics [134,136]. C_aw_NO and D_aw_NO can help to discriminate inflammation-related NO changes in the airways detected by an increase in C_aw_NO from changes in the bronchial wall caused by remodeling and detected by D_aw_NO changes [136].

Even smaller and cheaper instruments and single-use, disposable NO detectors are warranted to further widen access to FeNO measurement and cope with the medical need for the home monitoring of patients.

### 2.2. Exhaled Carbon Monoxide

The other gaso-transmitter, CO, is also present in exhaled breath and has two distinct sources: air pollution and tobacco smoke from the environment and the breakdown of heme to biliverdin from different cells in the body [17,22,137,138]. The reaction is catalyzed by heme oxygenase (HO) enzymes. The constitutive HO (HO-2) is an important transmitter in physiological conditions and has important neuroprotective properties and a role in male reproduction, while the inducible form (HO-1) is part of the antioxidant defense and has anti-inflammatory properties [139,140]. HO-1 is highly expressed in various inflammatory diseases, tumors and several other disorders [17,18,21,141,142,143,144,145,146]. Much of its effect is linked with the released CO that has widespread interactions with different cells and cell functions. Both HO-1 and HO-2 are identified as therapeutic targets in widely different diseases [147,148,149,150,151]. CO and carbon-monoxide-releasing molecules (CORM) are also investigated as therapeutic potentials [152,153].

CO can be detected in exhaled breath in a parts per million (10^−6^, ppm) range, and it is present with elevated levels in patients with inflammatory airway diseases such as asthma, cystic fibrosis (CF), obstructive sleep apnea and non-CF bronchiectasis [18,21,99,154,155,156,157]. The mean value of exhaled CO in healthy non-smoking subjects varied between 0.94 and 3.86 ppb in the 15 studies assessed in a meta-analysis [158]. In asthma, airway macrophages express higher levels of HO-1 together with elevated CO levels in exhaled breath than those from healthy people [18]. This observation together with other lines of evidence suggests that HO-1 is involved in the immune regulation of allergic airway inflammation and might have a cytoprotective role in asthma [159,160].

Smoke exposure causes an elevation in exhaled CO concentration that is strongly related to the number of cigarettes smoked by smokers, and therefore it is frequently used to test smoking status and the results of smoking cessation [22,161,162]. Changes in the exhaled CO level due to environmental smoke exposure requires special attention when it is interpreted as a potential biomarker of endogenously produced CO.

For exhaled CO measurement, small, portable, handheld devices have been developed. In most of the detectors, electrochemical sensors are used for CO measurement. They detect CO in the ppm range with a resolution of 1 ppm. More sensitive detectors may enable clinicians to detect changes in ppb levels and shed light on more subtle changes in endogenous CO formation [163,164,165]. Gas chromatographic analysis, infrared laser spectroscopic techniques (cavity ring-down spectroscopy and integrated cavity output spectroscopy) and compact laser spectroscopy detectors, used for fire detection or environmental CO detection, may serve as potential techniques to improve sensitivity. For human use, it is important that the sensor has no cross-reactivity to water vapor and/or volatile organic compounds abundantly present in exhaled breath. As good examples, semiconducting metal-oxide-based sensors were identified as potential options for breath tests because of their small size, good sensor response and real-time results. For exhaled CO measurements, knowledge generated in relation to FeNO has successfully been applied. Ghorbani R. et al. used mid-infrared tunable diode laser absorption spectroscopy and measured CO exhaled with constant flows at 60, 120 and 250 mL s^−1^ through the mouth and the nose followed by measurement at 120 mL s^−1^ after a 10 s breath hold. They used the trumpet model with axial diffusion. By using this method they were able to show that end-tidal CO was largely independent of ambient air CO and that airway CO was slightly higher than and related to ambient air CO [164].

### 2.3. Exhaled Hydrogen Sulfide

The third known gaso-transmitter is a sulfur species, H_2_S [19,20]. H_2_S is a poisonous, flammable gas with a specific odor of rotten eggs. In the human body, it is produced from cysteine by three different enzymes, cystathionine γ-lyase, cystathionine β-synthase and 3-mercaptopyruvate sulfur-transferase, in a wide range of cells. As well as its cellular production, interstitial microbiota also releases H_2_S, and bacterial production is its major source in the colon and the oral cavity. Hydrogen sulfide plays a variety of physiological and pathophysiological roles including regulatory and modulatory functions in vasodilation, neurotransmission, oxidative stress, apoptosis, innate and adaptive immune responses, cytoprotection, cell growth and oxygen sensing [19,166,167,168,169,170,171,172]. Its functions are strongly interrelated with processes involving reactive oxygen and nitrogen species with close connections between H_2_S and nitrergic and CO-mediated signaling [173,174]. Under pathophysiological conditions, either overproduction or underproduction of H_2_S can be harmful. Depending on its concentration and chemical composition of the surrounding environment it can have pro-inflammatory and anti-inflammatory roles [166]. There are several approaches to modify the production and level of H_2_S to treat widely different diseases including inflammatory disorders and cancers [175,176,177,178,179].

H_2_S is present in exhaled breath in the ppb range and can be detected by gas chromatography, mass spectrometry, different electrochemical sensors and by other sensitive techniques [169,180,181,182,183,184,185,186] (Table 2). Furthermore, the human nose can detect smell if sulfide is present in the ppb range [184,187]. The molecule is highly unstable, and this aspect needs to be taken into account when it is measured in different biological samples [188].

The number of studies investigating exhaled H_2_S in different diseases is very limited. One of the difficulties related to the use of exhaled H_2_S as a potential biomarker is its production by oral and intestinal microbiomes [188]. H_2_S produced by oral bacteria can directly contribute to its exhaled level, while H_2_S released by gut bacteria could contribute to its systemic level by crossing the gut–blood barrier. Large numbers of Gram-negative anaerobic bacteria produce volatile sulfur compounds (VSCs) including H_2_S during proteolytic degradation of proteins. To take the oral bacterial contribution into account, instead of collecting breath through oral sampling, it can be collected nasally. In the study of Liu N et al., the exhaled H_2_S level was 25.00 ± 17.94 ppb, and in nasally sampled breath, it was 10.59 ± 4.53 ppb in healthy subjects [189]. Nasal-exhaled H_2_S (NeH_2_S) was found to be less than 5 ppb in 90.6% of 1600 healthy subjects with an average value of 2 ppb in another study with a somewhat different sampling procedure [190].

**Table 2 micromachines-14-00391-t002:** Various hydrogen sulfide detection approaches in different phases of matter samples with potential clinical utility. Abbreviations: ppb—parts per billion; ppm—parts per million; VSC—volatile sulfur compounds; LOD—limit of detection; iv—intravenous; FMS—frequency modulation spectroscopy; 2*f*-WMS—second harmonic wavelength modulation spectroscopy; OA-ICOS—off-axis integrated cavity output spectroscopy; PAS—photoacoustic spectroscopy; ISE—ion sensitive electrodes.

Measurement Technique	Sample Type Used	Detection Range or Limit	Reference
Colorimetric detection of H_2_S using an etching-resistant effect on silver nanoprisms	Gaseous H_2_S released from garlic and Na_2_S solution in phosphate-buffered saline dilution series	Linear range from 1.03 to 32.9 μM H_2_S μM	Ahn, Y.J. et. al. [180]
Conclusion: Ag NPRs-coated H_2_S sensing paper demonstrated high selectivity, good sensitivity and good reproducibility and stability, together with a fast response time. Possible tool for on-site colorimetric detection of free H_2_S gas for exhaled breath analysis.			
Spectroscopic techniques for H_2_S measurement in gaseous mixes: FMS, 2f -WMS, OA-ICOS, PAS	VSC gas containing H_2_S	LOD from 500 ppb to 8.4 ppm	Ciaffoni, L. et. al. [181]
Conclusion: Laser-based spectroscopic sensors are possibilities for accurate breath diagnostics in clinical environment.			
Amperometric detection of H_2_S gas in N_2_ gaseous mix	10 ppm H_2_S in 99.95% pure N_2_ gas	Linear range from 75 ppb to 820 ppb	Gatty, H.K. et. al. [182]
Conclusion: The ppb-level detection capacity, combined fast response and high sensitivity to H_2_S makes the sensor potentially suitable for oral breath monitoring with a miniaturized handheld instrument.			
A paper-based fluorescent sensor for in situ gaseous H_2_S determination	Gaseous mix of H_2_S and purified air	LOD of 3 ppb	Petruci, J.F. et. al. [183]
Conclusion: An automated portable sensor for in situ determination of H_2_S gas that can be utilized in a clinical environment.			
Interscan RM-17 sulphide detectors	H_2_S gas exhaled from human subjects who received iv. Na_2_S solution.	LOD of 10 ppb–5 ppm	Toombs, C.F. et. al. [184]
Conclusion: The aim of the study was to prove that exhaled H_2_S represents a detectable route of elimination in the human body after iv. infusion of Na_2_S solution.			
Ion sensitive electrodes for H_2_S detection	Mammalian plasma	LOD of 100 nM, detection range: 1–100 μM	Xu, T. et. al. [185]
Conclusion: There is broad range of applications of the ISE-based H_2_S sensor, but the system requires high maintenance from the operator.			
Polarographic H_2_S sensors	Plasma or mammalian tissue homogenates	LOD from 2 μM down to 5 nM in certain methods	Xu, T. et. al. [185]
Conclusion: Compared to ISE-based H_2_S sensor, method detects H_2_S without an external electrical potential, has a simple structure, good reproducibility, a short response time and contains fewer noble metals. A great disadvantage of the liquid electrolyte sensors is that they leak easily, are prone to dry up and have a large residual current.			
Enzyme-based electrochemical H_2_S biosensors	Environmental water	LOD of 1 ppm in gas phase, LOD of 0.3 μM, linear response in the range of 1.09–16.3 μM in H_2_S solution	Xu, T. et. al. [185]
Conclusion: The enzyme-based H_2_S biosensor shows great advantages of selectivity and sensitivity. However, one of the most regrettable characteristics of the enzyme-inhibition-based biosensors is the different inhibition degrees caused by different inhibitors. Moreover, this kind of sensor also highly depends on the pH concentration due to its inhibition on the activity of enzyme, making it difficult to apply to in vivo H_2_S detection.			
Exhaled H_2_S on test paper with an ultrasensitive and time-gated luminescent probe	Breath exhaled by the mice	Semi-quantification of gaseous H_2_S in the range of 10–30 ppm	Zhang, R. et. al. [186]
Conclusion: The test paper imprinted by the complex probe ink can visualize clearly the trace H_2_S gas exhaled by the mouse.			

One study demonstrated that the exhaled hydrogen sulfide level is related to the type of airway inflammation in asthma. The exhaled H_2_S level was elevated in neutrophilic airway inflammation and negatively correlated with sputum eosinophil count [191]. In line with this, nasal-exhaled air was sampled, and the sulfide level tended to be lower in patients with allergic rhinitis, a typical eosinophilic disease of the upper airways, than in healthy subjects [190]. There are conflicting data published on the exhaled sulfide level in chronic obstructive pulmonary disease (COPD). One study showed that it was higher in COPD patients without sputum eosinophilia [192]. Exhaled sulfide was lower in COPD patients compared to healthy subjects and correlated with airway obstruction in another study [193]. Other authors found that exhaled H_2_S was correlated with FeNO but not with sputum inflammatory cells in COPD [194]. Furthermore, it was found that air pollution causes an increase in exhaled H_2_S concentration in patients with COPD, likely due to worsening airway inflammation [195]. Based on the above, exhaled sulfide level was suggested to be a potential biomarker of asthma and COPD [196]. Changes in exhaled sulfide levels can also be detected in sepsis, and different intestinal diseases including irritable bowel syndrome, colorectal adenoma, oral squamous cell carcinoma and chronic pancreatitis [189,197,198,199,200,201]. Due to the limited number of studies and different methodologies used, and the large contribution of oral bacteria to its exhaled level, further studies are needed to explore its potential role as a biomarker of asthma and COPD or other diseases.

## 3. Exhaled Hydrogen Peroxide

H_2_O_2_ is an oxygen metabolite that diffuses through cells and tissues and serves important metabolic and regulatory roles under physiological and pathophysiological circumstances. It is an important signaling molecule playing a part in cellular adaptation to environmental stress as a part of redox signaling pathways [202,203,204]. In oxidative stress and inflammation, NO, CO and H_2_S are interrelated with H_2_O_2_ and other reactive oxygen species in multiple ways [146,205,206,207,208,209]. Exhaled H_2_O_2_ can be captured in exhaled breath condensate (EBC), a cooled breath sample containing large numbers of volatile and non-volatile biomaterials [40,50]. The level of exhaled H_2_O_2_ is extremely variable and depends on several factors that having a direct or indirect influence on its level. Environmental conditions, ventilatory pattern, measurement techniques and storage influence its concentration directly, but they may also act indirectly by changing the pH of EBC [40,210,211,212,213,214,215]. To limit variability due to sample storage and support point-of-care detection, different online detection systems and disposable sensors have been built and tested [212,216,217,218,219,220,221].

To allow deeper understanding of oxidative-stress-related processes and interactions between different mediators, micromachines able to detect complete sets of molecules from the same sample are desirable.

## 4. Breathomics—Breath Fingerprinting

Different diseases have characteristic metabolic profiles that can be captured by using metabolomics, proteomics and other “omic” technologies in different biological samples. Using “omics” for biomarker discovery studies is one of the important pathways enabling us to reconstruct our understanding of different chronic diseases by measuring exhaled breath volatiles [222,223]. Thousands of different VOCs have been detected in exhaled breath that can be identified and quantified by mass-spectrometry-based methodologies or samples that can be discriminated based on the patterns by electronic or biological noses [2,35,36,37,38,224]. Exhaled VOCs are principally isoprene, alkanes, methylalkanes and benzene derivatives. They are related to widely different cellular functions and metabolic processes including lipid peroxidation, oxidative stress and cholesterol synthesis among others [5]. Endogenously produced VOCs can be detected in different samples, such as exhaled breath, urine, feces, saliva and blood. The concentration of a given VOC in exhaled breath is also influenced by alveolar minute ventilation and cardiac output together with its blood–gas partition coefficient. As well as endogenous formation, they can also be found in the environment or in other exogenous sources (food and drink, diagnostic test drugs, medication, smoking, etc.). VOCs found in biological samples cannot, therefore, solely reflect bodily functions because exogenous VOCs also have an influence on the exhaled samples. Discrimination between the two sources in exhaled breath samples is challenging and relies on using different breathing maneuvers, assessing the effect of VOC clean gases for inhalation, using filters in the inhalation loop of the sampling device and keeping a certain time gap between exposure and sampling (i.e., subject is requested not to smoke for 1–12 h before sample collection). A specific potential confounding source is the collecting device itself because several materials and most cleaning fluids release VOCs, and that is extremely hard to exclude. In general, environmental influence on the concentration of exhaled VOCs cannot be completely ruled out by any of the currently used approaches [40].

### 4.1. Spectrometry-Based Measurements

Gas chromatography mass spectrometry (GC-MS) allows precise identification and quantification of the molecules present in exhaled breath and is frequently used for unbiased metabolomics analysis [1,2,3,4]. The GC works on the principle that individual molecules can be separated from complex mixtures when heated. In GC, the mobile phase contains the breath sample in a gas state that is driven through a capillary column coated with a stationary phase (gas–solid chromatography) by inert gases (such as helium, nitrogen or synthetic air). The molecules of the mobile phase interact with the molecules in the stationary phase and depending on the compound’s retention time based on polarity and boiling point, compounds are eluted from the column at different timepoints and enter the MS. In MS, VOCs are ionized and fragmented using electronic or chemical ionization. Based on the mass-to-charge ratio (*m*/*z*) of the product ion, the original compound can be determined using library searches of the spectra and quantified based on the peak areas that are proportional to the quantity of the corresponding VOC. Due to their sensitivity, the specificity of GC-MS is considered as the “gold standard” in the measurement of the complex mixtures of compounds including exhaled breath volatomics and metabolomics [225,226]. Two-dimensional gas chromatography coupled with time-of-flight mass spectrometry (GCxGC-ToF-MS) has been proposed as an even more powerful tool for the multidimensional analysis of chemical mixtures. Phillips M. et al. demonstrated that by using this technique, more than 2000 molecules could be detected in breath samples, many more than detected earlier by one-dimensional techniques [227]. Other technologies such as selected ion flow tube mass spectrometry (SIFT–MS) and proton transfer reaction mass spectrometry (PTR–MS) have rapid response times, offering possibilities for online measurements of breath samples [228,229].

In exhaled breath samples, VOCs are usually present in very low concentrations, such as within the nanoM to picoM range (parts per billion–parts per trillion), making it difficult to measure them directly. To reach the detection limit of the available instruments, VOCs can be collected on sorbent traps and preconcentrated before separation by GC. For exhaled breath, different preconcentration methods, including multibed needle trap and solid-phase microextraction (SPME), have been used successfully [230,231].

Although some GC-MS systems are portable and can be applied in online measurements, the use of GC-MS as a POC test is limited by the complexity of measurement, the high-cost of equipment, the need for highly trained personnel for their operation and the challenges in the interpretation of obtained data.

Some developmental steps were taken to design a POC breath test that does not rely on MS. Hagens LA et al. designed a prototype of a small, easy-to-use and fast (<2 h from sample to result) POC breath test for octane measurement in breath samples from a ventilated patient in an intensive care unit (ICU). Their instrument relies on gas chromatography using air instead of helium as the carrier gas and a photoionization detector instead of MS. Octane was present in the samples in sub-ppb concentrations and the POC test was accurate in 95% of cases [232].

Technology has evolved in the last decades resulting in more sensitive spectrometers that are able to analyze breath samples without the need for pretreatment or GC separation, and direct mass spectrometry methods have become available.

Some direct mass spectrometry methods are readily usable for direct online analysis. Gas chromatography ion mobility spectrometry (IMS), may offer an opportunity for point-of-care use and methodology development to discriminate between environmental and endogenous VOCs [233,234]. Furthermore, secondary electrospray ionization high-resolution mass spectrometry (SESI-HRMS) and high-resolution mass spectrometry with direct analysis of a real-time ion source also offer real-time detection of the metabolome [235,236]. Developmental steps in detecting instruments were accompanied by new possibilities of data analysis by machine learning and artificial intelligence [237,238].

Due to the unmet need for POC breath tests, laser-spectroscopic-based breath analysis has gathered increasing attention. The mid-infrared (MIR) laser sources are good options for POC breath tests due to their high sensitivity, accuracy and reasonable prices. Several laser spectroscopic techniques have been studied and are under development including TDLAS, cavity ring-down spectroscopy (CRDS), photoacoustic spectroscopy (PAS) and quartz-enhanced photoacoustic spectroscopy (QEPAS) [126,239,240]. A combination of Fourier-transform infrared (FTIR) spectroscopy with artificial intelligence (AI) in a portable device has provided a convincing point-of-care mass screening of SARS-CoV2 infection [241] (Table 3).

The potential usefulness of the exhaled breath test in lung cancer screening, diagnosis, subtyping and personalized treatment is a very attractive field pointing towards a breathomics-supported diagnosis and treatment algorithm coupled with clinical parameters [242,243,244,245,246]. This might be of special interest in those challenging areas such as small-cell lung cancer and mesothelioma where progress is slower than in non-small-cell lung cancer. In children, an exhaled breath test offers a completely non-invasive mode of sampling, and exhaled biomarkers are expected to provide information previously inaccessible on ongoing airway inflammation in different airway diseases including asthma and cystic fibrosis [247,248].

**Table 3 micromachines-14-00391-t003:** Conventional and novel spectrometry-based detection approaches to identify VOC patterns with potential discriminative value in the diagnosis of pulmonary diseases. Abbreviations: BPN—benign pulmonary nodule; CF—cystic fibrosis; COPD—chronic obstructive pulmonary disease; FTIR—Fourier-transform infrared spectroscopy; GC/MS—gas chromatography/mass spectrometry; MPM—malignant pleural mesothelioma; NSCLC—non-small-cell lung cancer; TB—tuberculosis; ToF-MS—time-of-flight mass spectrometry; VOC—volatile organic compound.

Measurement Technique	No. of Detected Exhaled Compounds	Subjects	Findings	Reference
Computer-assisted GC/MS	150 peaks	Healthy (*n* = 17) and lung cancer patients (*n* = 14)	49 peaks differed between groups	Gordon SM. et al. 1985 [3]
Computer-assisted GC/MS	22 selected for further analysis	Patients undergoing bronchoscopy because of chest radiograph abnormalities (*n* = 108)	22 VOCs discriminated between patients with and without lung cancer	Phillips M. et al. 1999 [4]
Computer-assisted GC/MS	13 VOCs selected for further analysis	NSCLC (*n* = 36); COPD (*n* = 25); asymptomatic smoker (*n* = 35) and non-smokers (*n* = 50) controls	A logistic regression model using the concentration of the 13 VOCs classified 82.5% of subjects correctly	Poli D. et al. 2005 [243]
Computer-assisted GC/MS	12 selected VOCs were studied	CF patients with stable (*n* = 15) condition and exacerbation (*n* = 5); healthy controls (*n* = 20)	The alveolar gradient for pentane was higher in CF patients (with highest values in patients with pulmonary exacerbations) and inversely proportional to FEV1; (0.73 versus 0.24 ppb). CF patients exhibited a lower output of dimethyl sulphide	Barker M. et al. 2006 [247]
High-resolution MS	N.A.	Young non-smoking healthy adults (*n* = 10)	65 and 55 major compounds identified in positive and negative ion mode, respectively; diurnal changes in VOC spectra described	Xu L. et al. 2022 [236]
Multicapillary column/ion mobility spectrometer	N.A.	Patients with MPM (*n* = 52); healthy controls (*n* = 52) and asbestos workers without symptoms (*n* = 59) and benign asbestos-related diseases (*n* = 41); patients with non-asbestos-related lung diseases (*n* = 70) and lung cancer (*n* = 56)	Discrimination of patients with MPM from healthy, asymptomatic asbestos-exposed subjects, and from patients with other lung diseases and cancer with high specificity and sensitivity	Lamote K. et al. 2017 [233]
Multicapillary column/ion mobility spectrometer	35 of the peaks were identified in all subjects	Healthy subjects (*n* = 18). Simultaneous measurement of inhaled air and exhaled breath	Results facilitate the calculation of alveolar gradients and selection of truly endogenous VOCs	Westhoff M. et al. 2022 [234]
Fourier-transform infrared (FTIR) spectroscopy	N.A.	Emergency patients tested for SARS-CoV-2 (*n* = 297)	With the aid of an artificial intelligence algorithm, SARS-CoV-2 positivity is detected with high specificity and sensitivity based on exhaled breath samples	Shlomo I.B. et al. 2022 [241]
Selected ion flow tube mass spectrometry	116 specific human breath biomarker VOCs	Patients with lung cancer (*n* = 148), healthy controls (*n* = 168)	Based on VOC pattern, a prediction model of high accuracy was developed to predict lung cancer	Tsou P. H. et al. 2021 [245]
GC–ToF–MS	N.A.	Paediatric patients with persistent asthma (*n* = 96)	15 VOCs were selected as good predictors, and were used to build a prediction model that could discriminate between persistently controlled and uncontrolled asthma with high accuracy	Van Vliet D. et al. 2016 [248]
Proton transfer reaction ToF-MS	N.A.	Patients tested for SARS-CoV-2 infection: symptomatic positives (*n* = 270), asymptomatic positives (*n* = 27) and negatives (*n* = 840)	Based on VOC pattern, a prediction model of positivity is developed with specificity and sensitivity similar to conventional test methods	Liangou A. et al. 2021 [228]
High-pressure photon ionization ToF-MS	28 selected VOCs	Patients with lung cancer (*n* = 84 and 157), healthy controls (*n* = 368)	16 VOCs as lung cancer breath biomarkers were identified; including these 16 VOCs, a diagnostic model of high accuracy was developed	Wang P. et al. 2022 [246]
Thermal desorption coupled with GC/MS	N. A.	Patients with MPM (*n* = 14) and healthy controls (*n* = 20)	Ten VOCs were identified to be able to discriminate between MPM patients and healthy controls	Di Gilio A. et al. 2020 [244]
Thermal desorption coupled with GC/MS	6983 different VOCs observed in 352 subjects	Patients with lung cancer (*n* = 160), with BPN (*n* = 70) and healthy controls (*n* = 122)	19–20 VOCs discriminated lung cancer patients from healthy subjects and from patients with BPN	Chen X. et al. 2021 [242]

### 4.2. Electronic Noses

In contrast to spectrometry, electronic noses use a different approach and individual odorants cannot be determined exactly; only the sum of signals generated by the complex mixture of VOCs is categorized as different (or not) from the stored breathprints of control subjects [38,249]. They represent a portable, low-cost, easy point-of-care technique for pattern analysis of exhaled volatile compounds. These devices have composites of a sensor array together with a built-in processor and functionally resemble the biological olfaction. The other fundamental element of the electronic nose is the automated pattern recognition system, which uses computed mathematical algorithms and artificial intelligence to analyse and classify the detected signal pattern and requires powerful bioinformatics.

Changes in breathprints were detected due to pregnancy by e-nose and changes in VOCs during menstrual cycles were also detected by high-resolution real-time mass spectrometry suggesting a potential of breath tests to reflect female sexual hormone metabolome [250,251]. Several diseases have been characterized by distinct breathprints [252,253,254,255,256]. Rapidly evolving sensor technology and machine learning approaches have paved the way for wide use of electronic noses in exhaled breath studies in a variety of diseases (Table 4) [38,257].

There are various electronic nose detector systems available with a wide range of sensors including metal oxide, conducting polymer, nanomaterial-based or optical sensors.

Conducting polymer sensor arrays equip the Cyranose 320 device that is the most extensively used electronic nose in clinical practice and research [249,254]. The sensor films are loaded with conducting carbon black polymers. During measurement, the VOCs of the collected exhaled air are absorbed by the polymer. The consequential swelling of the polymer induces an increase in electrical resistance compared to the baseline, which can be detected as an electrical signal [258]. The polymers’ specific response to the absorbed molecule is based on its chemical characteristics (molecular size, shape, dipole moment, etc.) [258]. A mixture of VOCs produces a signal on the individual sensors that is the linear summation of responses to individual VOCs of the mixture weighed by their fractional amount absorbed by the polymer [258,259]. The sensor array consists of several cross-reactive sensors. The mathematical analysis of their signal patterns allows for the detection of subtle changes in the composition of the analyzed gas mixture [258,260]. Concerning their detection limits, these sensor types are moderately sensitive (with detection limits between 0.1 and 100 ppm. However, they have been shown to discriminate breathprints of various diseases with good sensitivity, such as lung cancer [261,262], acute respiratory distress syndrome [263], asthma [264,265], COPD [266], malignant mesothelioma [267], cystic fibrosis [268], etc. (Table 4).

Metal oxide semiconductive sensors detect the VOC molecules by reacting with them at high temperatures [249,269]. VOCs oxidize the metal oxide structure and thereby alter their conductivity, and this is detected as an electrical signal. The signal is related to the concentration of VOCs that reacts with the surface. Various metal oxides have been used as sensor surfaces in electronic noses, with tin oxide, tungsten oxide and titanium oxide being the most common [270,271,272]. The sensitivity of the surfaces can be enhanced by attaching palladium or gold nanoparticles to catalyze the reaction with VOCs [269], and also by the application of metal oxide nanowires to magnify the specific sensor surface area [271,273]. Their diagnostic application is not widespread. There have been a few studies with the “Diagnose” device to test its utility in human diagnostics in general and in ventilator-associated pneumonia [274] (Table 4). However, arrays of metal oxide sensors have been successfully tested in other settings, such as the food industry [269] and environmental applications [270].

Colorimetric sensors also offer a simple approach to design electronic nose detector systems. These sensors utilize dyes (e.g., metalloporphyrin dyes, pH indicators, molecules with large permanent dipoles), which respond to the binding of VOC molecules with a change in color. The color change can be detected with various colorimetric methods both qualitatively and quantitatively [249].

Moreover, quartz microbalance (QMB) gas sensors have been integrated in an electronic nose (LibraNose) that has been successfully tested in various clinical conditions [35,275,276,277] (Table 4). In these sensors, metalloporphyrin molecules are attached to QMB crystals. VOC binding to the metalloporphyrin surface modifies the oscillation of the QMB piezoelectric crystal, which can be detected as an electric signal.

A challenging approach in electronic nose development is to design custom-based sensor arrays applying nanomaterial-based technologies. Nanotechnology provides the opportunity to design individual sensors that are specific to certain molecules [278,279]. This allows for the tailoring of unique detector systems by including sensors in the array that are sensitive to VOC molecules, which have been identified with GC/MS techniques to have a good discriminative power in the diagnosis of certain diseases [249,280,281,282,283,284,285,286,287] (Table 4).

**Table 4 micromachines-14-00391-t004:** Representative examples for medical utilization of different electronic nose sensor systems.

Sensor Types (Detection System)	Diseases	References
Conducting polimer sensor arrays (Cyranose 320)	Lung cancer	[261,262]
	Acute respiratory distress syndrome	[263]
	Asthma	[264,265]
	COPD	[266]
	Malignant mesothelioma	[267,288]
	Cystic fibrosis	[268]
	Breast cancer	[253]
	Colorectal carcinoma	[289]
	Preeclampsia	[290]
Metal oxide semiconductive sensors (Diagnose)	Ventilator-associated pneumonia	[274]
Quartz microbalance sensors (Libranose)	Asthma	[275]
	Halitosis	[276]
	Lung cancer	[277]
Custom-designed arrays using nanomaterial technologies	Ovarian cancer	[280]
	Lung cancer	[281,282,287]
	Pulmonary arterial hypertension	[283]
	Multiple sclerosis	[284]
	Alzheimer’s disease	[285]
	Parkinson’s disease	[285]
	Pulmonary tuberculosis	[286]

Several methodological studies have addressed the details of breath sampling, sample storage and the reproducibility of measurements [14,58,59]. Physiological changes, such as exercise, have also been shown to cause a change in the breathprint detected by the e-nose, and this was confirmed by the changes in VOCs determined by mass spectrometry [60,291,292]. There is an expert consensus-based recommendation for technical standards to be applied when using e-noses for exhaled breath testing [293].

Among the methodological issues related to electronic nose technology, the collection of breath samples is of utmost relevance (Figure 2).

The expiratory flow rate should be standardized as it may influence the levels of various volatile organic compounds in the expired air as well as the electronic nose results. [261,294,295,296]. Moreover, it is recommended to perform exhalation against a 10–20 cm H_2_O resistance to close the nasal palate in order to minimize nasal contamination. Unfortunately, only a minority of studies available in the literature were carried out controlling for these circumstances. Another important issue is to avoid the contamination from ambient air [261,296], as it can highly influence the composition of exhaled volatiles. For this purpose, a VOC filter can be used to adsorb environmental volatiles during inspiration. The exclusion of dead space would also be required because it dilutes and modifies the VOC content of the alveolar air [261,296]. Moreover, the water vapor pressure and temperature of the exhaled breath may also influence the precision of the measurements in the case of some sensors [249]. The material of collection bags and the duration and circumstances of the storage and transportation of samples are also not negligible factors and may influence measurement results. Concerns regarding sample collection and handling have been thoroughly reviewed by Bikov A et al. [294].

## 5. Conclusions

In summary, various sampling and analytical methods have been used to assess the metabolome through exhaled breath. While individual gaso-transmitters paved the way for clinically useful point-of-care measurements, currently, the rapid development in sensor technology and the application of artificial intelligence have resulted in major developments in the field of breathomics, a promising field for easy-to-use, point-of-care machines for diagnostic and monitoring purposes. Advanced wearable sensors to detect biomolecules in fluids or exhaled breath open a way for potential online home monitoring [297]. The main areas of interest are screening, diagnosis, phenotyping, exacerbation prediction, exacerbation etiology and prediction of the treatment response where a major breakthrough can be achieved with the envisioned micromachines.

## Figures and Tables

**Figure 1 micromachines-14-00391-f001:**
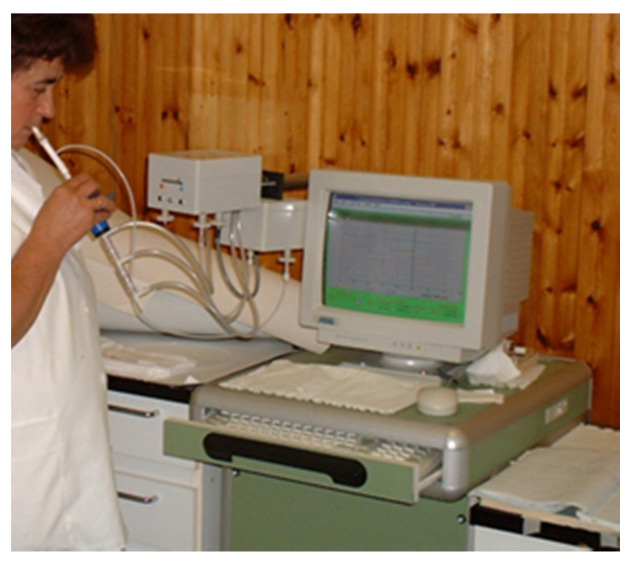
One of the original chemiluminescence analyzers (Logan model LR2500, Logan Research; Rochester, Kent, UK) for online measurement of exhaled NO in the early nineties. The analyzer is sensitive to NO from 1 ppb (by volume) to 5000 ppb with a resolution of 0.3 ppb. It also measures CO_2_ (resolution 0.1% CO_2_; response time, 200 ms) in real time. The subject exhales slowly (5–6 L/min) from total lung capacity for 15–20 s against resistance to exclude nasal contamination. During expiration the pressure is kept constant (3 ± 0.4 mm Hg) by using a visual display of expiratory flow measured by pressure and volume sensors in the analyzer.

**Figure 2 micromachines-14-00391-f002:**
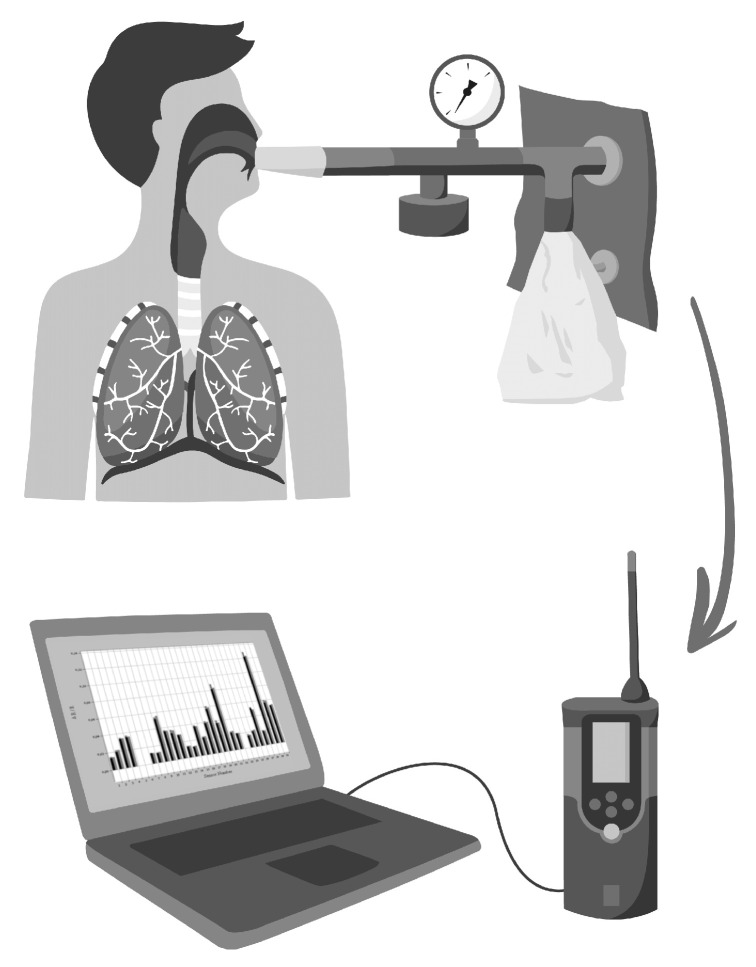
The recommended collection of breath samples. The subject inhales ambient air through a VOC filter, and exhales against a resistance at a controlled flow rate. The first fraction of exhaled air is collected in a separate bag to exclude dead space air from the sample. Sample is collected in a special collection bag and is analyzed with the electronic nose device.

## Data Availability

Not applicable.

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
