# Peer review of "Exhaled Biomarkers for Point-of-Care Diagnosis: Recent Advances and New Challenges in Breathomics"

_micromachines, 2023, doi:10.3390/mi14020391_

Round 1

Reviewer 1 Report

The review paper entitled “Exhaled biomarkers for point of care diagnosis: recent advances and new challenges in breathomics” is an interesting and comprehensive study on the topic of exhaled gaso-transmitters in medical practice.

I would suggest some minor improvements:

1. The introduction may be additionally divided by subheadings, because it is quite wide and presents many different information.

2. The Authors underline their willingness to present the technical and theoretical problems but with strong connection with clinical practice. Nevertheless, for example the part dedicated to FeNO gives very few information about clinical application. Therefore, I would expect to add the summary of clinical application of FeNO method and to list (for example as the additional table) the papers trying to use FeNO in different diseases. There are for example studies on FeNO in eosinophilic esophagitis and inflammatory bowel disease. In current version of the manuscript there is almost only about bronchial asthma.

3. In medicine, the most frequently used methods in analyzing exhaled gases are hydrogen and methane breath tests. I think there is to mention about it, even if the Authors are focused on “breathonomics” and spectrometry-based measurements.

4. Writing about H2S the Authors very briefly write about possible production of sulfite gass also by oral cavity bacteria, which is very important in terms of methodology of assessment this particle as a biomarker of different diseases. There is variability of oral microbiome and there is to underline, that the specific microbiome may lead to false positive results of breath tests not correlated with real diseases. There is to name some of the types of bacteria metabolizing with peculiar gas particles production.

5. Writing about medical practical application of “electronic noses” I would just mention attempts to screen with this method for cancers other than the only mentioned lung cancer (there are for example studies on colon cancer). Interesting would be to add few words about studies with the animals – the dogs and their noses are the model to follow.

Author Response

Dear Reviewer,

Thank you for your comments. Please see our responses in the attached document.

Reviewer 2 Report

The authors nicely describe methods of sampling and analysis of metabolites through exhaled air. They briefly describe their usefulness in diagnosing and assessing the activity of various diseases. They aim to provide an overview of recent advances and challenges in the measurement of respiratory biomarkers, with an emphasis on the utility of their measurement as a non-invasive diagnostic and on-site monitor. The article contains a complete detailed description of the methods introduced so far, perhaps too extensively. I would suggest that it would be good to define their usefulness in diseases a bit more broadly.

Author Response

Dear Reviewer,

Thank you for your critical reading of our manuscript and for your positive evaluation. We are also grateful for the helpful comment, in which he/she suggested to “define the usefulness of the described methods in diseases a bit more broadly”.

We agree that more emphasis should put on the diseases  and conditions in which the potential diagnostic application of breath analysis has been studied. Therefore, we added 2 additional tables to the manuscript. In Table 1 we demonstrated selected relevant studies which investigated the application of FeNO testing in different diseases and conditions, whereas in Table 4. we summarized relevant studies which used eNose techniques to identify diseased individuals based on breath VOC pattern.

Furthermore we elaborated with more details on clinical utility of gaso-transmitters, as well. 

We look forward your feed-back. 

Ildiko Horvath

corresponding author on behalf of the authors

Reviewer 3 Report

The manuscript "Exhaled biomarkers for point of care diagnosis: recent advances and new challenges in breathomics" shows new diagnostic techniques for diseases. However, the authors must address the following point:

In each of the sections, the relationship of the transmitting gases (NO, CO and H2S) by the human body with chronic diseases such as cancer is not clear. The authors must detail the relationship of these gases through the pharmacological targets involved and adding values that may be comparable with normal situations.

Author Response

Dera Reviewer,

We appreciate your time and effort for your critical reading of our manuscript and helpful comment to address a relevant aspect, which was not emphasized strongly in the manuscript. With respect to your comment and suggestion, we have made  changes in the manuscript 

and added parts to all three gaso-transmitter providing more details on their role in health and disease. Please, see lines 172-180; 219-241; 246-249; 342-350; 390-395; 400-402; 415-419. Our review does not aim at explaining the pathophysiological processes linked with these molecules, that is a very broad and exciting field of medicine, but focuses on their detection in exhaled breath. Therefore we kept the pathomechanical aspects relatively short. We also provided information on levels of exhaled NO, CO and H2S in healthy subjects. 

Looking forward to your feed-back,

Sincerelly yours,

Ildiko Horvath

corresponding author on behalf of all authors